# Hobnail Papillary Thyroid Carcinoma, A Systematic Review and Meta-Analysis

**DOI:** 10.3390/cancers14112785

**Published:** 2022-06-03

**Authors:** Ariadni Spyroglou, Georgios Kostopoulos, Sofia Tseleni, Konstantinos Toulis, Konstantinos Bramis, George Mastorakos, Manousos Konstadoulakis, Kyriakos Vamvakidis, Krystallenia I. Alexandraki

**Affiliations:** 12nd Department of Surgery, Aretaieio Hospital, National and Kapodistrian University of Athens, 11528 Athens, Greece; aspyroglou@gmail.com (A.S.); kbramis@gmail.com (K.B.); mastorakg@gmail.com (G.M.); konstadoulakismm@yahoo.com (M.K.); 2Department of Endocrinology, 424 General Military Hospital, 56429 Thessaloniki, Greece; ge0k1990@gmail.com (G.K.); touliskos@gmail.com (K.T.); 3Department of Pathology, Medical School, University of Athens, 11527 Athens, Greece; stseleni@med.uoa.gr; 4Department of Endocrine Surgery, Henry Dunant Hospital Center, 11526 Athens, Greece; info@drvamvakidis.gr

**Keywords:** papillary thyroid carcinoma, hobnail, mortality

## Abstract

**Simple Summary:**

Although papillary thyroid carcinoma (PTC) has, in general, a very good prognosis, aggressive variants of this tumor present worse clinicopathological characteristics and a worse clinical outcome. One of the most recently identified aggressive variants is the hobnail PTC. We herein summarize the clinicopathological characteristics of all so far reported cases along with our small case series with this histological characteristic and investigate the correlation of its presence with the clinical outcome in these patients and their mortality. We, furthermore, analyze various clinicopathological characteristics such as age, tumor size, presence of distant or lymph node metastases, extrathyroidal extension, and lymphovascular invasion, and their impact on mortality. In conclusion, as this variant negatively affects mortality, its presence should be carefully assessed in all papillary thyroid carcinoma cases.

**Abstract:**

Although papillary thyroid carcinoma (PTC) is considered to have an excellent prognosis, some recently identified more aggressive variants show reduced overall survival rates. Hobnail PTC (HPTC) was newly recognized as one of these aggressive forms, affecting recurrence, metastasis, and overall survival rates. Herein, we performed a systematic review and meta-analysis of studies including cases or case series with patients with HPTC. Furthermore, we included our individual case series consisting of six patients. The pooled mortality rate in the cohort, consisting of 290 patients, was 3.57 (95% CI 1.67–7.65) per 100 person/years. No sex differences could be observed concerning mortality (*p* = 0.62), but older age and tumor size significantly affected mortality (*p* = 0.004 and *p* = 0.02, respectively). The percentage of hobnail cells did not affect mortality (*p* = 0.97), neither did the presence of *BRAF* mutations. Classical characteristics such as the presence of extrathyroidal extension (*p* = 0.001), distant metastases (*p* < 0.001), and lymph node metastases (*p* < 0.001) all had a significant impact on mortality. Thus, HPTC appears to correlate with worse overall survival, and all PTC cases should be carefully assessed for this variant.

## 1. Introduction

According to the Surveillance, Epidemiology, and End Results (SEER) program, thyroid cancer is the most common endocrine malignancy, representing 2.3% of all new cancer cases in the United States, with female patients being affected almost three times more often than males [1,2]. Among the various histological types, papillary thyroid carcinoma (PTC) is the most common type, accounting for 70–80% of all cases. Although PTC is considered to have an excellent prognosis, with a 10-year relative survival rate of 98% in cases with localized disease, some histological variants of PTC have shown a more aggressive profile, with reduced overall survival rates [3]. These aggressive subtypes include the tall cell (TC), columnar cell (CC), and hobnail subtype (HPTC) [4]. These correlate with higher rates of recurrence and metastasis and in some cases with refractoriness to radioiodine treatment [5,6,7]. Further PTC subtypes include the diffuse sclerosing (DS), solid/trabecular (ST), clear-cell (CLC). In DS, *BRAF* mutations are rather uncommon events, whereas *RET/PTC* rearrangements is one of the main genetic alterations observed. In ST, *RET/PTC* alterations have been documented in pediatric patients upon radiation exposure but not in adults, whereas TC and to a lesser extent CC and ST present *BRAF* mutations. HPTC presents, aside from *BRAF* mutations, *TP53* mutations and occasionally *NOTCH1* mutations [5]. The correct diagnosis of these aggressive variants presents a diagnostic challenge but also plays a significant role for the risk stratification and most importantly for the therapeutic approach in these patients. 

Recently, HPTC was identified as one of these aggressive subtypes. HPTC is characterized by high nuclear/cytoplasmic ratio, apically placed nuclei, and loss of cellular polarity, presenting a hobnail and micropapillary pattern [8,9,10]. Asioli et al. defined initial diagnostic criteria for this thyroid carcinoma subtype, including a non-solid type of PTC, with tall/columnar or diffuse sclerosing features in <10% of the tumor and >30% of the neoplastic cells presenting hobnail features [10]. Some subsequent studies documented an aggressive profile with worse prognosis, also in tumors with 10-30% hobnail features, also correlating the presence of this feature with a higher rate of lymph node and distant metastases, and higher mortality [11,12,13,14,15,16].

Thus, the presence of hobnail features in PTCs appears to negatively affect survival but whether the percentage of hobnail burden correlates with worse outcome still remains unclear. Herein we performed a systematic review and meta-analysis of the existing literature to assess the mortality rate in patients with HPTC, either as sole histological diagnosis or in combination with other histological subtypes, and to investigate the interrelation of various clinicopathological characteristics in these tumors. 

## 2. Materials and Methods

For the present systematic review, the PubMed, Scopus, and Cochrane Databases were searched for relevant studies for inclusion. The search included studies published from inception of the database to 7th March 2021. This review was conducted according to the PRISMA guidelines (PROSPERO ID CRD42022300923). Search keywords included: hobnail, micropapillary, discohesive, papillary thyroid cancer/carcinoma/neoplasm/tumor. References were then uploaded to EndNote X9 (Clarivate, Philadelphia, PA, USA) and screened for relevance. Articles were critically appraised to assess the risk of bias. An overall judgement of the methodological quality of the eligible studies was based on the following criteria: selection of participants, ascertainment of exposure and outcome, causality, and reporting.

Case reports and case series with any proportion of HPTC were included in the present study, whereas non-human studies, reviews, non-English or French studies, and studies lacking demographic or histopathological data were excluded. Data represented in two different studies were only included once. Two reviewers (AS and KIA) independently screened titles and abstracts of all studies fulfilling the inclusion criteria and remaining full texts were then screened in detail by both reviewers. In cases of disagreement, a third reviewer (KT) determined eligibility. 

For each study included in the present analysis, first author, publication year, study characteristics, and patient demographics were registered. Extracted data included: sex, age, tumor size (mm), percentage of tumor cells with hobnail features, lymphovascular invasion (LVI), extrathyroidal extension (ETE), histological combinations (with: classical PTC or poorly-differentiated carcinoma/anaplastic thyroid carcinoma, or TS, or CC, or follicular variant (FV)), *BRAF* status (wild-type (WT), or mutant (MUT)), distant metastases (DM), lymph node metastases (LN), radioiodine treatment (RAI), recurrence, follow-up duration (alive no disease (AND), or alive with disease (AWD), or dead of disease (DOD) or dead of other causes (DOC) in months). In addition to the case series collected through literature search, the respective data of our own case series (Aretaieion Hospital, Athens, “Henry Dunant” Hospital Center, Athens, 424 General Military Hospital, Thessaloniki; 363/13-10-2021) consisting of six patients with HPTC were included in the meta-analysis according to previous reports suggesting that combination of an original observational study with a systematic review is an appropriate approach in low incidence disease entities [17,18]. For the inclusion, strict diagnostic criteria, in line with the most recent pathological definition of HPTC [19], were applied upon examination of the six cases by an experienced thyroid pathologist in order to distinguish “hobnail-like” features seen in classic PTC versus the “true HPTC”. 

All analyses were weighted according to the number of patients affected and performed using R Project for Statistical Computing (version 3.6.3) (The R Foundation for Statistical Computing, Vienna, Austria). The primary outcome was aggregate mortality rate in patients with HPTC to incorporate follow-up time in the analysis. Mortality rate was defined as the number of patients with HPTC who died of the disease divided by the total number of person-time at risk. The total number of person-months at risk, as reported in the primary studies, was used, and for comparison with the existing literature, was finally converted to person-years. In case these numbers were not reported, the average follow-up time and the total number of patients with HPTC were used to approximate the total person-time at risk [20]. We performed a random-effects (RE) meta-analysis using a generalized mixed model (GLMM) to pool the primary studies [21]. GLMM was preferred, as it accounts for within-study variance and has better performance in small datasets, compared to traditional meta-analytic methods [22,23]. I^2^ and τ^2^ were calculated to estimate the between-study variance. Heterogeneity was further assessed by calculating a 95% prediction interval (95%PI), as proposed by Higgins [24]. A 95% PI provides an estimation of the outcome measure (mortality rate) in a new study [24]. Moreover, Egger’s test and visual inspection of funnel plot were applied to assess for potential publication bias. To conduct the meta-analysis, the “*metarate”* function from the “*meta”* package was used. Furthermore, we performed secondary exploratory analyses investigating possible relationships between various clinicopathological features and the mortality rate. In the exploratory analyses, studies reporting the respective patient-level data were included. Group comparisons were performed either with a T-test and Wilcox test (continuous variables) or Chi-square and Fisher’s Exact Test (categorical variables). In accordance with existing literature, a cut-off of 30% hobnail features was used in our analyses [10]. In addition, we also presented a multivariable logistic regression model for mortality. The choice of variables was based on published literature and clinical judgement [25]. Level of statistical significance was set at 5% (*p* < 0.05) for all statistical tests.

## 3. Results

### 3.1. Descriptive Characteristics of the Included Studies

A total of 601 articles were identified through our literature search and, after duplicates removal, 301 articles remained. After screening by title and abstract, 47 articles were left for full text review. Out of these, 29 articles fulfilled the criteria and could be included in our systematic review [11,12,13,14,15,16,19,26,27,28,29,30,31,32,33,34,35,36,37,38,39,40,41,42,43,44,45,46,47] (Figure 1). 

The present analysis includes 290 patients (284 patients from the eligible case studies and 6 patients from our own case series, Table 1 and Figure 2) with presence of HPTC. Out of these, 185 were female (63.8%) and 105 (36.2%) were male. Mean age at diagnosis was 51.3 years (range 14–92) and mean follow-up time was 42.6 months (range 2–274). The mean size of the tumors was 29.8 mm (range 5–90 mm) and the presence of >30% hobnail component in the tumors was 58.9%, out of the 263 patients who had available data. In 125 patients (62.5%), HPTC co-existed with a classic PTC, in 46 patients (23%) with at least two further histological types (poorly differentiated (PDC) or anaplastic thyroid carcinoma (ATC), FV, TC, CC, ST, or CLC). Extrathyroidal extension was present in 101 patients (48.8%) and lymphovascular invasion in 155 (58.9%). A *BRAF* V600E mutation was identified in 150 examined cases (71.1%), and 134 patients (78.4%) received radioiodine treatment upon thyroidectomy. A recurrence was documented in 47 patients (28%). At the end of the respective follow-up period, 31 patients had died (14.8%) due to their disease. Almost half of the patients (135) presented lymph node metastases, whereas 43 (25.7%) presented distant metastases. The descriptive characteristics of the included studies can be found in Table 1.

### 3.2. Primary Meta-Analysis 

The aim of the present study was to investigate the mortality rate in patients with HPTC. Out of the 29 articles included in the present systematic review, 23 could also be included in the meta-analysis. A funnel plot was generated to assess publication bias for these publications, and the Egger’s test was not statistically significant for asymmetry (*p* = 0.35, Figure 3), indicative of a low publication bias. The pooled mortality rate was 3.57 (95% confidence interval (CI) 1.67–7.65) per 100 person/years, with a prediction interval of 0.37 to 34.30 (Figure 4).

### 3.3. Exploratory Analyses

In addition to the primary meta-analysis, further exploratory analyses investigating the correlation between various patients’ characteristics and mortality were performed. No sex differences could be observed concerning mortality (male odds ratio (OR) 1.25, 95% CI 0.52–3.00, *p* = 0.62). Unlike sex, age significantly affected mortality, with alive patients being significantly younger (median 50.5 years old (36.2–63)) than patients dead of their disease (median 62.5 years old (51.5–67.8), *p* = 0.004). Tumor size also played a significant role on mortality, as longer survival correlated with patients with smaller tumors (23 mm, 16.5–39.5) in comparison to patients with larger tumors (42 mm, 25–50, *p* = 0.02). The percentage of hobnail cells did not affect mortality, and tumors with >30% hobnail cells did not present increased likelihood of mortality (OR 0.99, 95% CI 0.41–2.38, *p* = 0.97). Similarly, the absence of *BRAF* mutations did not improve mortality (OR 1.61, 95% CI 0.30–8.52, *p* = 0.67). However, the presence of extrathyroidal extension (OR 24.97, 95% CI 1.36–457.1, *p* = 0.001), of distant metastases (OR 24.17, 95% CI 7.74–75.5, *p* < 0.001) and of lymph node metastases (OR 20.94, 95% CI 2.73–160.72, *p* < 0.001) all had a significant impact on mortality. RAI did not exert a significant effect on mortality (OR 1.15, 95% CI 0.36–3.67, *p* = 1).

In the multivariable model to assess mortality in combination with various clinicopathological characteristics, the size of the tumor, and the percentage of hobnail cells were selected, due to the limited number of events. According to this analysis, for each millimeter increase in the tumor size, mortality increased by 3%, adjusted for the percentage of hobnail cells (*p* = 0.032), (Table 2).

## 4. Discussion

The aim of the present study was to systematically review the existing literature on the rather newly identified aggressive variant of PTC, namely HPTC, and to assess its impact on mortality and its interrelation with further clinicopathological characteristics of these patients. Our analysis demonstrated a mortality rate of 3.57 per 100 patient-years, much higher than the thyroid cancer incidence-based mortality of 0.46 per 100,000 patient-years obtained through the SEER-9 cancer registry program for 2010–2013 [48]. In our meta-analysis, a small number of events could be documented, which could also explain the rather large heterogeneity of the analysis, shown here by the wide prediction interval. The overall mortality in our meta-analysis was 14.8%, whereas a further large multicenter study investigating mortality in PTC observed a mortality of 1.1% and 5.3% in PTC patients without or with a *BRAF* mutation, respectively [49]. Similarly, in a Japanese study including 5897 PTC patients, with a median follow-up period of 177 months, overall mortality was 7%, with the PTC-related mortality reaching only 2% [3]. The HPTC overall mortality of 14.8% in our study does not significantly differ from a previous review of the, at the time, existing literature performed by Lee et al., in 2015, with an overall mortality of 19% [37]. Unlike that, Watutantrige-Fernando et al. observed an overall mortality of 0% in their HPTC cohort [14]. Likewise, our own case series consisting of six patients presented an overall mortality of 0%, with a follow-up range of 5–106 months. This large heterogeneity in the mortality of the HPTC could be attributed to the variation in the follow-up periods or the diagnostic criteria applied for the respective definition of HPTC, as the cases included herein refer to timely different WHO classifications of endocrine tumors. Furthermore, the variety of histological combinations, and in particular the co-existence of HPTC with PDC or ATC, could explain the heterogeneity in the mortality calculated. 

The female predominance in PTC with an approximate ratio of 2:1 is well acknowledged in several studies and was also confirmed in our meta-analysis concerning HPTC [50,51]. However, although several studies showed that both all-cause and disease-specific mortality are more increased in male patients with PTC, in our study, no gender differences in mortality were observed in patients with HPTC [50,52,53]. In patients with classic PTC, mean age at diagnosis was found to be 48 years, whereas in our cohort of HPTC the mean age was higher, i.e., 51.3 years [48]. Older age is a known negative predictive factor for survival, and this was also confirmed in the case of HPTC, with older age correlating with reduced survival [54,55]. 

Mean tumor size for thyroid carcinoma is 19 mm, whereas a much larger size (29.8 mm) was found in HPTC [56]. Furthermore, as in classic PTC, here also, in HPTC, larger tumor size was associated with a worse survival [3]. Other clinicopathological characteristics such as lymph node metastases, extrathyroidal extension, and distant metastases are already well-acknowledged factors aggravating survival in thyroid cancer patients. In a large classic PTC cohort, lymph node metastases occurred in 19% of all patients included in the study, whereas in the present study 51.9% of all patients presented with lymph node metastases at diagnosis. Similarly, in classic PTC, extrathyroidal extension was observed in 13% of the patients, whereas we documented an ETE of 48.8%. Finally, synchronous metastases are present in 1% of classic PTC cases, whereas in cases with HPTC this percentage, including, though, both synchronous and metachronous metastases, increases to 25.7% [3]. As expected, in HPTC all these three characteristics were proven as negative prognostic factors, leading to increased mortality. 

It is widely recognized that *BRAF* mutations correlate with locally aggressive disease. In our study, 71.1% of all tumors with the HPTC were carrying *BRAF* mutations, a percentage much higher than the 45.7% observed in a large multicenter study of patients with classic PTC, accentuating the risk of HPTC to associate also with worse survival [49]. 

A significant role in mortality was previously attributed to the percentage of hobnail features in PTC. One of the first studies investigating this aggressive variant of PTC suggested the cut-off of 30% as differentiative of prognosis and found that tumors with >30% of hobnail features were associated with increased mortality [11]. Unlike that, a later study applying the same cut-off did not identify any differences in mortality, distant metastases, or lymph node involvement [14]. In our meta-analysis, including 290 patients, the percentage of hobnail cells does not additionally affect mortality, possibly suggesting that, alone, the presence or absence of HPTC might be crucial for survival. In our multivariable analysis, presence of >30% hobnail cells was not associated with increased mortality adjusted for tumor size. A previous meta-analysis, including, though, only 124 HPTC patients, documented an increased lymph node metastasis rate in patients with >30% hobnail cells, but found no significant differences in the rates of distant metastases and further clinicopathological characteristics known to affect mortality [25]. 

Taken together, these findings highlight the aggressive behavior of HPTC, with larger tumors, frequent extrathyroidal extension, lymph node and distant metastases and increased mortality. In particular, as the presence of HPTC is only identified postoperatively, its clinical consequences are difficult to prevent. Although the cut-off of 30% hobnail features is currently applied to establish the diagnosis, any documentation of hobnail features in histology should be considered as alerting to the physician with regards to the prognosis and management plan. Close monitoring of these patients could be important for their outcome, at least until more knowledge about this variant and its therapeutic options and pitfalls is achieved. 

## 5. Limitations

Limitations of this study include the small number of available studies in the field, the small number of cases included in each study, rendering some of them as low-level of evidence, the large variation on the clinicopathological data reported herein, the possibly less strict histopathological diagnostic criteria used by some of the studies included in the metanalysis (high interobserver variability even among thyroid experts), the differences in the follow-up periods and, thus, in the reported outcomes. 

## 6. Conclusions

In summary, the present systematic review shows that HPTC is associated with aggressive clinicopathological features, i.e., a larger tumor size at diagnosis, the presence of extrathyroidal extension, of lymph node metastases, of distant metastases, and an increased mortality. In comparison to the respective rates reported for classic PTC in the literature, the HPTC presents worse rates in all clinicopathological features associated with an aggressive phenotype and worse prognosis. Further studies should elucidate the impact of the hobnail cell percentage on prognosis and therapeutic strategies. 

## Figures and Tables

**Figure 1 cancers-14-02785-f001:**
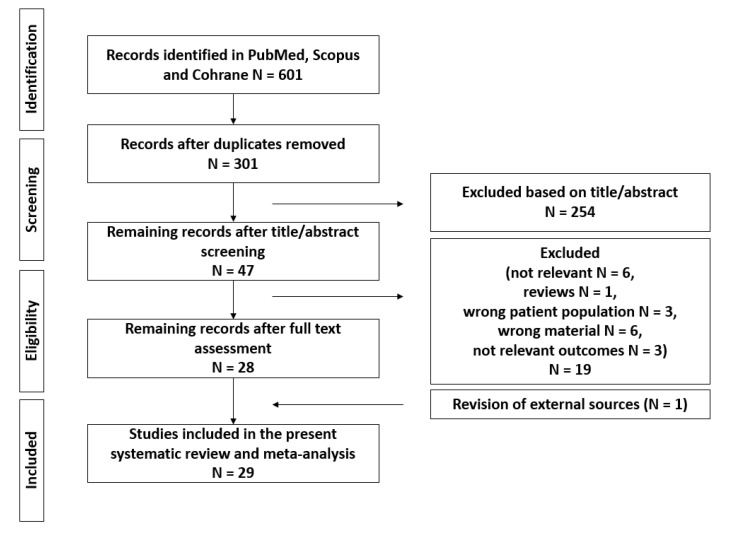
Flowchart for data collection according to PRISMA guidelines.

**Figure 2 cancers-14-02785-f002:**
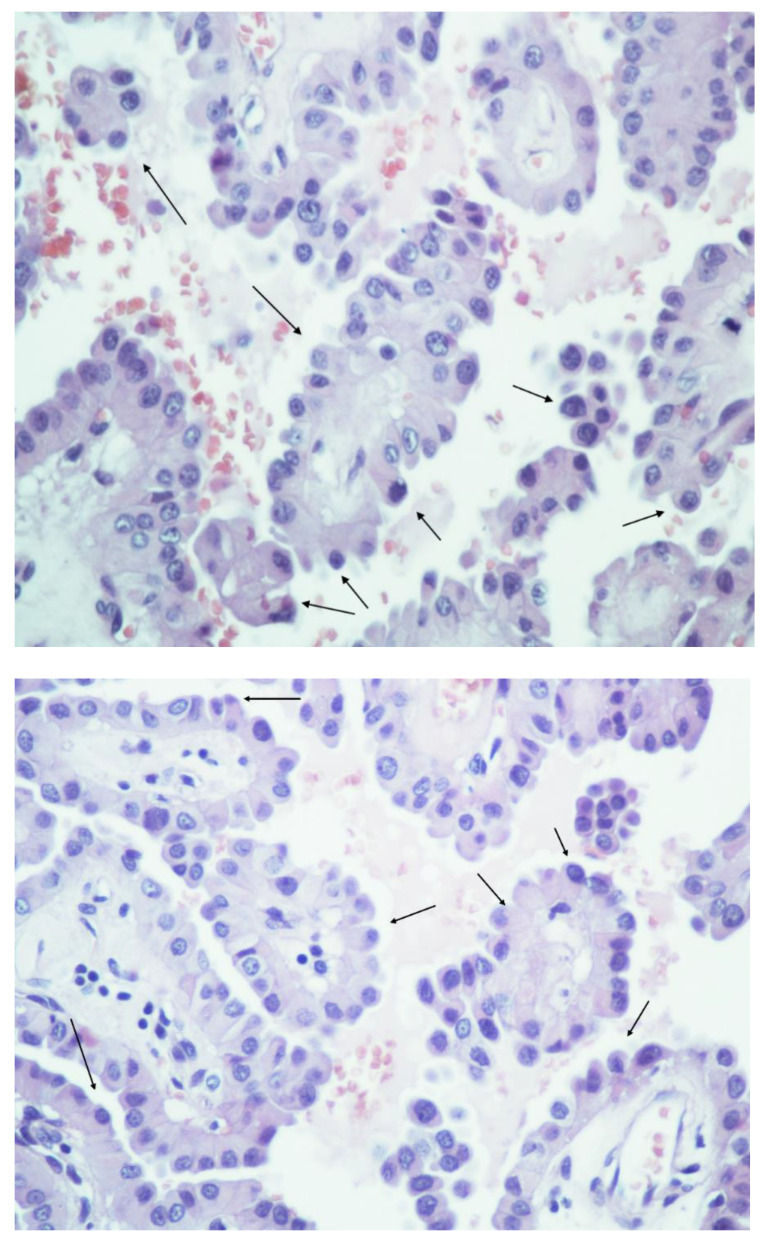
Examples of hobnail papillary thyroid carcinoma (HPTC) with presence of micropapillary structures, hobnail features in the cells with apically placed nuclei, loss of cellular cohesion (HE × 400). Arrows indicate hobnail cells.

**Figure 3 cancers-14-02785-f003:**
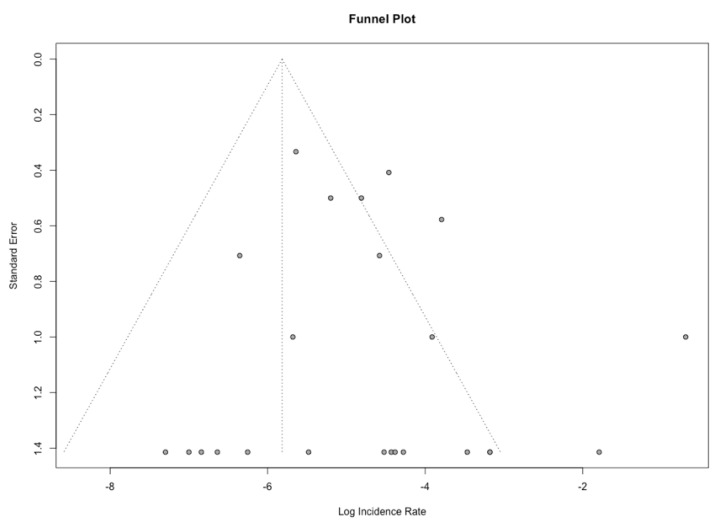
Funnel plot for the assessment of publication bias.

**Figure 4 cancers-14-02785-f004:**
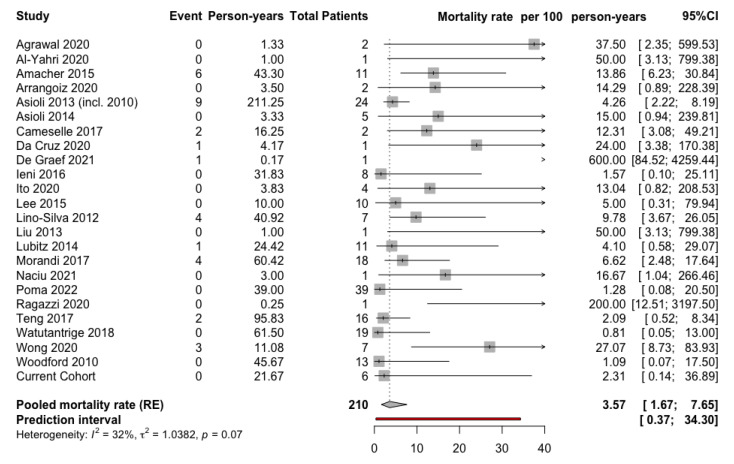
Forest plot demonstrating the mortality rate (in years) in patients with hobnail papillary thyroid carcinoma (HPTC). CI: confidence interval, *RE:* random effects model, *p:* refers to Cochran’s Q test for heterogeneity [11,12,13,14,15,16,19,27,28,29,31,32,33,34,35,36,37,39,40,41,42,43,47].

**Table 1 cancers-14-02785-t001:** Demographic and clinicopathological characteristics of included studies.

Study	N	Sex (m:f)	Age Range	Size Range (mm)	Hc >30%	Ete	Lvi	Histological Combinations	Fu (Months Range)	Outcome	*BRAF* mut.	Dm	Ln
Agrawal 2020 [27]	2	1:1	22–68	22–35	1/2	0/2	0/2	PTC	6–10	2/2 AND	2/2	0/2	0/2
Al-Yahri 2020 [28]	1	0:1	61	20	NA	0/1	0/1	NA	12	1/1 AND	1/1	0/1	0/1
Aliyev 2020 [26]	1	1:0	75	19	1/1	1/1	1/1	PTC, FV	NA	NA	NA	NA	1/1
Amacher 2015 [15]	11	7:4	22–76	9–65	2/6	4/5	NA	PTC, PDC, ATC, TC, CC	3.6–228	3/11 AND, 2/11 AWD, 6/11 DOD	2/4	4/7	10/10
Arrangoiz 2020 [29]	2	1:1	31–33	25–27	1/2	NA	NA	PTC	18–24	2/2 AND	NA	0/2	NA
Asioli 2013 [11]	24	6:18	28–78	10–70	12/24	NA	17/24	NA	4–274	12/24 AND, 3/24 AWD, 8/24 DOD, 1/24 DOC	NA	10/23	13/24
Asioli 2014 [12]	5	2:3	27–86	20–90	2/5	NA	4/5	PTC, FV	2–24	4/5 AND, 1/5 AWD	3/5	0/5	3/5
Bellevicine 2012 [30]	1	1:0	57	20	NA	NA	NA	PTC	NA	NA	1/1	NA	NA
Cameselle-Teijeiro 2017 [31]	2	1:1	53–62	17–65	2/2	2/2	2/2	PTC, PDC, FV, ST	67–128	2/2 DOD	1/2	2/2	2/2
Da Cruz 2020 [32]	1	1:0	67	49	1/1	NA	1/1	PDC, ATC	50	1/1 DOD	NA	1/1	1/1
De Graef 2021 [33]	1	0:1	38	entire gland	1/1	1/1	NA	PTC, FV, ST	2	1/1 DOD	1/1	1/1	1/1
Ieni 2016 [34]	8	2:6	47–69	10–32	8/8	NA	4/8	NA	39–60	8/8 AND	3/8	0/8	2/8
Ito 2020 [35]	4	0:4	70–79	19–42	4/4	2/2	2/2	PTC, TC	5–18	3/4 AND, 1/4 AWD	NA	1/4	4/4
Lee 2015 [37]	10	4:6	32–68	6–40	10/10	7/10	8/10	PTC, FV, TC	9–28	9/10 AND, 1/10 AWD	8/10	0/10	8/10
Lilo 2017 [38]	1	1:0	81	14	1/1	1/1	1/1	NA	NA	NA	NA	NA	NA
Lino-Silva 2012 [13]	7	4:3	27–68	39–50	0/7	NA	5/7	NA	63–84	3/7 AWD, 4/7 DOD	NA	3/7	5/7
Liu 2013 [36]	1	0:1	17	32	1/1	1/1	1/1	PTC, FV, ST	12	1/1 AWD	NA	0/1	1/1
Lubitz 2014 [39]	12	3:9	21–80	5–65	10/10	7/12	5/12	PTC, TC, ATC	12–57	7/11 AND, 3/11 AWD, 1/11 DOD	8/10	3/12	10/12
Mehrotra 2019 [44]	1	0:1	66	53	1/1	0/1	1/1	PTC, FV, TC	NA	NA	NA	0/1	1/1
Morandi 2017 [40]	18	4:14	3 < 45, 15 > 45	12 < 40, 6 > 40	18/18	16/18	18/18	NA	7–128	7/18 AND, 7/18 AWD, 4/18 DOD	12/18	8/18	NA
Motosugi 2009 [45]	1	1:0	57	55	1/1	1/1	0/1		NA	NA	NA	0/1	0/1
Naciu 2021 [41]	1	1:0	47	entire gland	1/1	1/1	NA	PTC	36	1/1 AND	1/1	0/1	0/1
Poma 2022 [47]	99	41:58	49.8 (15.9) *	18 (13–24) **	34/99	40/99	49/99	PTC, TCV, ST, CLC	12	19/39 AND, 20/39 AWD	69/88	NA	32/99
Ragazzi 2020 [16]	1	0:1	71	50	0/1	NA	NA	PTC, ATC	3	1/1 AWD	1/1	1/1	1/1
Schwock 2015 [46]	1	0:1	26	21	1/1	NA	NA	PTC, TC	NA	NA	NA	NA	1/1
Teng 2017 [42]	18	5:13	23–78	10–50	18/18	6/18	2/18	PTC, TCV, CC	12–101	13/16 AND, 1/16 AWD, 1/16 DOD, 1/16 DOC	16/17	2/18	10/17
Watutantrige-Fernando 2018 [14]	25	10:15	24–73	7–80	16/25	NA	24/25	NA	13–67	13/19 AND, 6/19 AWD	14/24	3/10	17/24
Wong 2020 [19]	7	2:5	48–92	25–80	6/7	5/7	NA	PTC, ATC, TC	median 23	3/7 AND, 1/7 AWD, 3/7 DOD	3/3	3/7	6/7
Woodford 2010 [43]	18	2:16	14–86	NA	NA	2/18	8/18	PTC, FV	3–166	12/13 AND, 1/13 AWD	0/9	0/18	1/13
current study	6	5:1	31–53	11–50	1/5	3/6	2/5	PTC, PDC, TC, FV	5–106	2/5 AND, 3/5 AWD	2/5	1/5	3/5
Summary of all studies	290	105 (36%): 185 (64%)	14–92 (51.3)	5–90 (29.8)	155 (59%)	101 (49%)	155 (59%)	PTC 125 (63%), 2 + S 46 (23%), PDC/ATC 4 (2%), TC 18 (9%), CLC 1 (0.5%), NOS 2 (1%)	2–274 (42.6)	AND 121 (58%), AWD 55 (26%), DOD 31 (14.8%), DOC < 0.1%	150 (71%)	43 (26%)	135 (52%)

N: Total number of cases per study included, HC: Hobnail cells, ETE: extrathyroidal extension, LVI: lymphovascular invasion, FU: follow-up, DM: distant metastases, LN: lymphnode metastases, PTC: papillary thyroid carcinoma, AND: alive no disease, NA: not available, FV: follicular variant, PDC: poorly differentiated carcinoma, ATC: anaplastic thyroid carcinoma, TC: tall cell subtype, CC: columnar cell subtype, AWD: alive with disease, DOD: dead of disease, DOC: dead of other causes, CLC: clear cell subtype, 2 + S: two or more subtypes, NOS: no other subtypes, * Reported as mean (standard deviation), ** Reported as median (interquartile range).

**Table 2 cancers-14-02785-t002:** Multivariable analysis for the outcome mortality. Two parameters (i.e., size and HPTC %) were selected, due to the number of deaths of 29, considering 10 events per parameter. OR: odds ratio, CI: confidence interval, HPTC: hobnail papillary thyroid carcinoma.

Characteristic	OR	95% CI	*p*-Value
Size	1.03	1.00, 1.06	0.032
HPTC 30% No	-	-	-
HPTC 30% Yes	0.76	0.26, 2.28	0.6

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
