# Peer review of "Hobnail Papillary Thyroid Carcinoma, A Systematic Review and Meta-Analysis"

_cancers, 2022, doi:10.3390/cancers14112785_

Round 1

Reviewer 1 Report

Spyroglou and colleagues performed a metanalysis on HVPTC to assess mortality rate and prevalence of clinicopathological features in this aggressive variant of PTC.

A similar article was published in 2021 by Donaldson and collaborators (DOI: 10.1007/s12020-020-02505-z), and the present study do not add significant information on this topic. Indeed, the study of Spyroglou includes 8 recent studies published after the publication of the metanalysis by Donaldson, which account for a total of 12 patients. In addition, an original series of 6 patients is included. Besides my concern about the novelty of the study, please find below my comments aimed at improving the manuscript.

Major.

1) The authors should consider including the largest series on HVPTC, which was published in 2022 (DOI: 10.3389/fendo.2022.842424).

2) The mortality rate reported by the authors is 18.2%, which significantly higher from that of Donaldson et al. (i.e., 10%). The large amount of case reports could introduce a substantial bias since they are low-quality data for metanalyses. In addition, a significant number of studies included HVPTC in combination with poorly differentiated and anaplastic carcinomas. In my opinion, this represents a significant bias when assessing the mortality rate, since the presence of less differentiated lesions deeply impact on the outcome. The authors should consider removing these lesions or assessing the mortality rate with and without less differentiated tumors.

3) The main focus of the study is the mortality rate, which is presented in terms of person-years. More details about the significance as well as the computing of this parameter should be provided by the authors.

4) The studies included in the metanalysis refer to different WHO classification of endocrine tumors. How was this managed by the authors? Particular attention should be dedicated to the 30% cut-off introduced in 2017.

5) In Figure 1, details about the records excluded after full-text screening should be reported as per PRISMA guidelines.

6) In Figure 3, more details should be reported. Which is the meaning of each column? There are two columns named “Events”, but I think one of these reports the weighted value. How is time expressed? In addition, what is the meaning of the p-value? Is it a Cochrane test?

Minor.

1) Please add arrows in Figure 2 to highlight hobnail cells.

2) In the introduction, the authors stated that “DSV and SV frequently carry BRAF or RET/PTC alterations”. Actually, BRAF mutation are rarely found in DSV as reported in the 2017 WHO classification of endocrine tumors. Also, RET/PTC were associated with SV in pediatric tumors of radiation-exposed cases; in the adult population this association was not confirmed.

3) In the discussion, it is reported that “BRAF mutations correlate with more aggressive features and increased mortality in PTC patients”. BRAF mutations are associated with locally aggressive disease, not with increased mortality. Even the independent impact of TERT promoter mutations on mortality is still debated.

4) Please always provide raw numbers along with percentages.

5) Please spell out abbreviation on first mention including in legends.

6) In Table 1, the total number of cases per study should be reported.

Author Response

Reviewer 1

Spyroglou and colleagues performed a metanalysis on HVPTC to assess mortality rate and prevalence of clinicopathological features in this aggressive variant of PTC.

A similar article was published in 2021 by Donaldson and collaborators (DOI: 10.1007/s12020-020-02505-z), and the present study do not add significant information on this topic. Indeed, the study of Spyroglou includes 8 recent studies published after the publication of the metanalysis by Donaldson, which account for a total of 12 patients. In addition, an original series of 6 patients is included. Besides my concern about the novelty of the study, please find below my comments aimed at improving the manuscript.

We would like to thank the reviewer for his constructive comments. Aim of our study was to perform a meta-analysis of a rare type of a common disease, mainly focusing on the mortality rate of HVPTC. In the revised version of our manuscript, according to the reviewer’s suggestion, the very recent, and so far, largest cohort study on HV has been included, as new evidence emerged, increasing the impact of the meta-analysis (lines 78-82).

Major.

1) The authors should consider including the largest series on HVPTC, which was published in 2022 (DOI: 10.3389/fendo.2022.842424).

We are thankful to the reviewer for pointing out this large case-series of patients with HVPTC, which was not included in our meta-analysis, since the time frame of our literature search ended the 7th of March 2021 along with the PROSPERO registration. According to the reviewer’s recommendation, this study is now included in the revised version of our systematic review and meta-analysis (lines 78-82: “For the present systematic review, the PubMed, Scopus and Cochrane Databases were searched, for relevant studies for inclusion. The search included studies published from inception of the database to 7th March 2021. This review was conducted according to the PRISMA guidelines (PROSPERO ID CRD42022300923).”). All results are updated accordingly (see text: Results section: lines 138-141, 144-159, 171-177 and 185-199 and Table 1, Figure 1, 3 and 4).

2) The mortality rate reported by the authors is 18.2%, which significantly higher from that of Donaldson et al. (i.e., 10%). The large amount of case reports could introduce a substantial bias since they are low-quality data for metanalyses. In addition, a significant number of studies included HVPTC in combination with poorly differentiated and anaplastic carcinomas. In my opinion, this represents a significant bias when assessing the mortality rate, since the presence of less differentiated lesions deeply impact on the outcome. The authors should consider removing these lesions or assessing the mortality rate with and without less differentiated tumors.

We fully understand the reviewer’s concern. The primary endpoint of our meta-analysis is the aggregate mortality rate expressed in 100 person-years, as the primary studies report different follow-up time. The overall mortality of initially 18.2%, now in the revised version 14.8%, reflects the ratio of patients with HVPTC who died of the disease (DOD) divided by the total number of the identified cases and is a descriptive statistical measure. We absolutely recognize that the overall mortality is higher compared to the one reported in a previous study by Donadson et al., which excluded patients with PDC and ATC, and, as the reviewer suggested, we also calculated the pooled mortality rate excluding these cases. Interestingly, though, we did not find large differences in the pooled mortality rate (without PDC/ATC cases: 3.17 (95%CI: 1.61-6.23) per 100-person years, vs. with PDC/ATC cases: 3.57 (95%CI 1.67-7.65)). Similarly, the percentage of patients DOD excluding those with PDC, or ATC was 12.7% (vs. DOD % including cases with PDC and ATC: 14.8%). Primary aim of our study was to investigate the mortality rate in HVPTC either alone, or in combination with other histological subtypes, providing important information to the reader (lines 73-76: “Herein we performed a systematic review and meta-analysis of the existing literature to assess the mortality rate in patients with HV, either as sole histological diagnosis or in combination with other histological subtypes, and to investigate the interrelation of various clinicopathological characteristics in these tumors.”). Nevertheless, we have added a comment regarding the possible effect of PDC/ATC in the analysis that cannot be neglected and has now been acknowledged in the Discussion of the revised version of our manuscript (lines 231-232: “Furthermore, the variety of histological combinations, and in particular the co-existence of HV with PDC or ATC could explain the heterogeneity in the mortality calculated.”).

3) The main focus of the study is the mortality rate, which is presented in terms of person-years. More details about the significance as well as the computing of this parameter should be provided by the authors.

We would like to thank the reviewer for this comment. The paragraph describing the statistical analysis has now been rephrased, in order to shed light to this complex calculation. The revised manuscript now reads: “The primary outcome was aggregate mortality rate in patients with HV of PTC to incorporate follow-up time in the analysis. Mortality rate was defined as the number of patients with HV of PTC who died of the disease divided by the total number of person-time at risk. The total number of person-months at risk, as reported in the primary studies, was used, and for comparison with the existing literature, was finally converted to person-years. In case these numbers were not reported, the average follow-up time and the total number of patients with HV were used to approximate the total person-time at risk.” (lines 111-118).

4) The studies included in the metanalysis refer to different WHO classification of endocrine tumors. How was this managed by the authors? Particular attention should be dedicated to the 30% cut-off introduced in 2017.

We fully agree with the reviewer, but since the 30% cut-off was a rather arbitrary definition, we have tried to shed light on the importance of a cut-off level or just the presence of HV of PTC. Nevertheless, this important limitation pointed out by the reviewer is now acknowledged in the revised version of the manuscript (lines 227-230: “This large heterogeneity in the mortality of the HV of PTC could be attributed to the variation in the follow-up periods or the diagnostic criteria applied for the respective definition of HV, as the cases included herein, refer to timely different WHO classifications of endocrine tumors.”).

5) In Figure 1, details about the records excluded after full-text screening should be reported as per PRISMA guidelines.

As the reviewer recommended, according to the PRISMA guidelines, details about the excluded studies are now reported in Figure 1 (Excluded (not relevant n=6, reviews n=1, wrong patient population n=3, wrong material n=6, no relevant outcomes n=3) N=19).

6) In Figure 3, more details should be reported. Which is the meaning of each column? There are two columns named “Events”, but I think one of these reports the weighted value. How is time expressed? In addition, what is the meaning of the p-value? Is it a Cochrane test?

According to the reviewer’s suggestion, Figure 4 has been modified. Time is expressed as person-years and p-value refers to Cochran’s Q test for heterogeneity (Figure legend, lines 181-183: “Figure 4: Forest plot demonstrating the mortality rate (in years) in patients with hobnail variant (HV) of papillary thyroid carcinoma (PTC). CI: Confidence Interval, RE: Random effects model, p: refers to Cochran’s Q test for heterogeneity.”)

Minor.

1) Please add arrows in Figure 2 to highlight hobnail cells.

Figure 2 has been modified according to the reviewer’s suggestion.

2) In the introduction, the authors stated that “DSV and SV frequently carry BRAF or RET/PTC alterations”. Actually, BRAF mutation are rarely found in DSV as reported in the 2017 WHO classification of endocrine tumors. Also, RET/PTC were associated with SV in pediatric tumors of radiation-exposed cases; in the adult population this association was not confirmed.

We would like to thank the reviewer for this remark. This statement has been corrected in the revised version of the manuscript (lines 54-58: “In DSV BRAF mutations are rather uncommon events, whereas RET/PTC rearrangements is one of the main genetic alterations observed. In SV, RET/PTC alterations have been documented in pediatric patients upon radiation exposure but not in adults, whereas TCV and to a lesser extent CCV and SV present BRAF mutations. HV present besides from BRAF mutations TP53 mutations and occasionally NOTCH1 mutations.”)

3) In the discussion, it is reported that “BRAF mutations correlate with more aggressive features and increased mortality in PTC patients”. BRAF mutations are associated with locally aggressive disease, not with increased mortality. Even the independent impact of TERT promoter mutations on mortality is still debated.

Here again we would like to thank the reviewer for his comment. This misleading statement has been removed from the revised version of the manuscript. The revised version of the manuscript reads: “It is widely recognized that BRAF mutations correlate with locally aggressive disease.” (line 255)

4) Please always provide raw numbers along with percentages.

As the reviewer recommended, raw numbers have been included in both text (lines 144-159) and Table 1 in the revised version of the manuscript.

5) Please spell out abbreviation on first mention including in legends.

As suggested, abbreviations are now spelled out on first mention, also in figure legends.

6) In Table 1, the total number of cases per study should be reported.

According to the reviewer’s suggestion, the total number of cases per study is now reported in the revised version of the manuscript (Table 1, second column (N)).

Reviewer 2 Report

A very well written article, limitations of the study were completely described.

I think that authors can add some can discuss better clinical consequences in relation to the fact that diagnosis usually came out postoperatively.

Author Response

Reviewer 2

A very well written article, limitations of the study were completely described.

I think that authors can add some can discuss better clinical consequences in relation to the fact that diagnosis usually came out postoperatively.

We would like to thank the reviewer for his comment. As suggested, the impact of the only post-operative diagnosis of the HV is now discussed in the revised version of the manuscript (lines 274-275: “In particular, as the presence of the HV is only identified postoperatively, its clinical consequences are difficult to prevent.”).

Round 2

Reviewer 1 Report

The authors have addressed all my comments.

In my opinion, the manuscript has been substantially improved.

Author Response

We would like to thank the reviewer for his comments.